# Microbial Melanin: Renewable Feedstock and Emerging Applications in Food-Related Systems

**Erminta Tsouko** †, **Eirini Tolia** † **and Dimitris Sarris** *

Laboratory of Physico-Chemical & Biotechnological Valorization of Food By-Products, Department of Food Science & Nutrition, School of the Environment, University of the Aegean, Leoforos Dimokratias 66, 81400 Myrina, Lemnos, Greece
* Correspondence: dsarris@aegean.gr
† These authors contributed equally to this work.

**Abstract:** Melanin is among the most important natural pigments produced by various organisms, from microbes to plants and mammals. Melanins possess great properties such as radioprotective and antioxidant activity, heavy metal chelation and absorption of organic compounds. The biosynthesis of melanin through the DOPA metabolic pathway and/or the DHN pathway mainly involves the tyrosinase and laccase enzymes that catalyze the oxidation of phenolic and indolic substrates to form melanin classes, namely eumelanin, pheomelanin, allomelanins and pyomelanin. The cost-efficient production of melanin at a large scale, with a chemically specified composition, constitutes a major technical challenge. Alternative production routes including highly efficient microbial stains cultivated on renewable resources could sustain and up-scale melanin production capacity. The strategy of valorizing low-cost and abundant agro-industrial waste and byproduct streams complies with concepts of sustainable development and circular economy, thus eliminating the environmental footprint. Genetic engineering tools could substantially contribute to enhancing melanogenesis in natural producers via target gene overexpression and the recombination of novel strains. The production of biobased films for food packaging applications reinforced with melanin nanoparticles constitutes a market segment of high interest due to environmental and societal concerns around the end-of-life management of conventional plastics, gradual depletion of fossil resources, sustainability issues and high performance.

**Keywords:** eumelanin; pheomelanin; allomelanins; pyomelanin; neuromelanin; food packaging; agro-industrial waste; overexpression; fermentation





## 1. Introduction

Melanin is a heterogeneous pigment that arises from the oxidation of phenolic and indolic molecules followed by the sequential polymerization of intermediate phenols and resulting quinones [1]. Organisms such as animals, plants, fungi, yeasts, bacteria, and protozoa produce melanin to survive and adapt to extreme environments such as exposure to toxic metals, osmotic shock and electromagnetic radiation [2]. There are five main categories of melanin based on the chemical configuration of the polymer, namely eumelanin, pheomelanin, neuromelanin, allomelanin and pyomelanin (Table 1). Most commonly, melanins range from brown to black coloration, while reddish and yellowish color can also be found, depending on the melanin type [3]. The most abundant melanins in nature are eumelanin, neuromelanin and pheomelanin [4]. The biosynthesis of allomelanin and pyomelanin is carried out though the oxidative polymerization of phenolic compounds, i.e., catechol and 1,8-dihydroxynaphthalene (DHN). The precursor molecules for eumelanin and pheomelanin are L-3,4-dihydroxyphenylalanine (DOPA), and cysteinyl-DOPA respectively. Routes for synthetic melanin production include the oxidation of dopamine or DOPA [5].

**Table 1.** Types of melanin and melanin producers.

| Type | Organisms | Color | Oxidating Substrate | References |
|---|---|---|---|---|
| Eumelanin | Animal, human, bacteria, fungi | Black—Brown | L-tyrosine | [6] |
| Pheomelanin | Animal (hair, feathers) | Yellow—Red | L-tyrosine and L-cysteine | [6,7] |
| Neuromelanin | Animal | Black—Brown | | [4] |
| Allomelanin | Plants, fungi, bacteria | Black—Brown | Catechol, caffeic acid, dihydroxynaphthalene, tetrahydroxynaphthalene, protocatechualdehyde, 4-hydroxyphenylacetic acid, γ-glutaminyl-4-hydroxybenzene | |
| Catechol melanin | Plants | Black—Brown | | [5] |
| DHN-melanin | Fungi, bacteria | Black—Brown | | |
| Pyomelanin | Fungi, bacteria | Black—Brown | Homogentisic acid | [8] |

Properties of melanin such as high refractive index, wide band absorption spectra (i.e., ultraviolet, visible and infrared), insolubility in conventional solvents, physicochemical stability radical scavenging ability, chelation for metal ion extraction, and high antioxidant activity render melanin a next-generation material with a polydiverse pool of applications in fields of cosmetics, pharmaceuticals, coloration, electronics, bioremediation and nanocomposites [1,9].

The worldwide melanin market size was valued at USD 13.7 million in 2022, while it is projected to reach USD 18 million by 2028, at a CAGR of 4.55% [10]. Natural sources of melanin include cuttlefish ink, black sheep wool, feathers of wild turkey and crow, black oat, black garlic, yeasts (*Cryptococcus neoformans*), fungi (*Aspergillus fumigatus*) and mushrooms (*Pleurotus cystidiosus* and *Armillaria cepistipes*) [2,5,9]. Commercial melanin-based formulations of either natural or synthetic origin are still expensive [9]. Alternative production routes involving highly efficient microbial stains cultivated on renewable resources could lead to sustainable and up-scaled melanin production capacity. Numerous agro-residues—fruit and vegetable waste, whey, molasses, corn steep liquor, and wheat bran—have been evaluated as potential nutrient sources to produce pigments of microbial origin [11,12]. The bioconversion of waste streams into added-value products such as pigments [13], biopolymers [14], organic acids [15,16], and microbial oil [17,18] comply with the circular bioeconomy era. Microbial biotechnology is the most important tool for the development of closed-loop circular bioprocesses that could stepwise re-introduce waste into the value-added food chain as novel products [19].

This review presents recent advances made towards the production of melanin. Fermentative approaches are discussed while strategies that comply with circular bioeconomy and sustainability issues are emphasized. More specifically, this review includes all the studies that have so far dealt with the valorization of renewable resources to produce nutrient-rich fermentation media for melanin production. Emerging applications of melanin in high value-added sectors are included while melanin's physicochemical characterization is reported and related to the aforementioned. Innovative food packaging formulations are also highlighted. Biosynthetic pathways for melanin production from fungal and bacterial strains focusing on the most important enzymes are presented. This review can serve as a guide that gives insight into bioprocessing-related pathways to enhance melanin production, followed by advanced valorization of melanin in numerous end uses. In this way, this compound could gain extra added value while new disposal markets can be proposed and/or discovered.

## 2. Microbial Pigments

The color of products in food-, pharmaceutical-, textile-, and cosmetic-based formulations affects the consumer preference. Recently, there have been several issues related to the adverse health effects of synthetic and artificial colors. Several synthetic colorants are derived from precursors that are fossil based and rationally non-renewable [12]. The

European Food and Safety Authority (EFSA) has set more strict thresholds regarding the acceptable dairy intake of colorants, namely quinoline yellow (E104), sunset yellow (E110) and ponceau 4R (E124). The industry seeks innovative pigment sources to replace synthetic sources since they have been related to allergenicity, carcinogenic and teratogenic activity [11]. Alternative natural pigments derived from plants and microorganisms as colorants could provide safe and green-labeled products of high quality and could satisfy increased health consciousness among consumers [20]. It should be highlighted that plant-derived pigments are directly dependent on climate changes [11]. Microbial pigments are biodegradable and non-toxic to humans [21] as well as biocompatible; further, they present strain-dependent versatility and flexibility, higher production rates, and straightforward handling of genes of bacteria and fungi [22]. In the pharmaceutical industry, microbial pigments have been found to contribute to the treatment of various diseases such as cancer, leukemia, and diabetes mellitus. Moreover, they have antibiotic and immunosuppressive properties [22]. Some of them have been approved by the FDA for use in food and they are a primary choice for industries since they offer visual appeal as well as antioxidant [23] and probiotic properties [22]. However, the selection of microorganisms for the production of value-added products must be performed carefully. Microorganisms that are used for food colorant production must be primarily non-pathogenic and non-toxic in their natural habitat. In particular, filamentous fungi should be studied for any toxins that may be pathogenic to the human body. The use of pigments produced by such microorganisms in food, farm and cosmetic products may cause serious health problems [21].

The production of microbial pigments such as melanins, carotenoids, flavins, and quinones (Figure 1) has become increasingly important in recent years while numerous microbial strains including bacteria, yeasts, fungi, algae and protozoa have been reported as potential bio-colorant producers [23]. The mostly applied microorganisms belong to species of *Monascus*, *Paecilomyces*, *Serratia*, *Streptomyces*, *Cordyceps*, *Rhodotorula*, *Cryptococcus*, *Pleurotus*, *Armillaria*, *Colletotrichum* and *Aspergillus* [22]. *Rhodotorula* yeasts are well known for their enhanced carotenogenic ability and they present great industrial value due to carotenoids production at a marketable scale [24]. Among fungal cultures, *Monascus* strains are most commonly employed for yellow, orange, and red pigment production via solid-state fermentation (SSF) while pigments have been used as natural coloring agents and food additives [25]. Black yeasts of the order *Chaetothyriales*, i.e., *Exophiala* genus, can produce notable melanin concentrations, while they present polymorphic proliferation patterns and enhanced capability to resist in extreme temperatures and radiation (non- and ionizing), and high salinity [26]. Several *Penicillium* strains have been reported to produce pigments, while only *P. oxalicum* is used for the industrial production of food-grade pigments [27].

The utilization of cost-effective carbon, nitrogen, and other nutrient sources derived from renewable resources could improve the fermentation efficiency of microbial pigment production. The fermentation media represent approximately 30–70% of the total production cost of bioprocesses [12,28]. Agri-food-based waste and byproduct streams are rich in valuable compounds, i.e., pectins, proteins, polyphenols, and oil. Their recovery is necessary for the subsequent valorization of the remaining residues that could serve as low cost, abundant and rich in fermentable sugars feedstock for microbial bioconversion [29]. Low-cost by-products and residues of agro-industrial effluents such as cheese whey, tomato waste, corn meal, coconut residue, peanut meal, soybean meal, sugarcane bagasse, fruit and vegetable waste, crude glycerol, coffee husks, bakery waste, brewery wastewater, corn cob, and grape waste have been investigated for the production of various bio-pigments aiming to reduce the bioprocess costs and optimize the pigment production. Depending on the microbial strain, SSF and submerged fermentation strategies have been employed [12,22]. Pretreatment strategies constitute a basic step prior to cellulose conversion to assimilable sugars, since the action of specific enzymes to hydrolyze target compounds is facilitated. Additionally, the stage of pretreatment is among the most expensive steps for the conversion of lignocellulosic biomass to fermentable sugars [18]. So far, several efforts have been made to render the process for bio-based pigment production more efficient and cost

effective. For instance, physical pretreatment strategies, i.e., milling and sieving, have been used to produce fermentation feedstock using sugarcane bagasse, onion peels, palm fronds and fruits and vegetable wastes. Other studies have employed chemical methods such as diluted acid pretreatment or alkaline treatment prior to enzymatic hydrolysis with commercial enzymes or enzymes produced by the fungal strains *Aspergillus awamori* and *Aspergillus* [12].

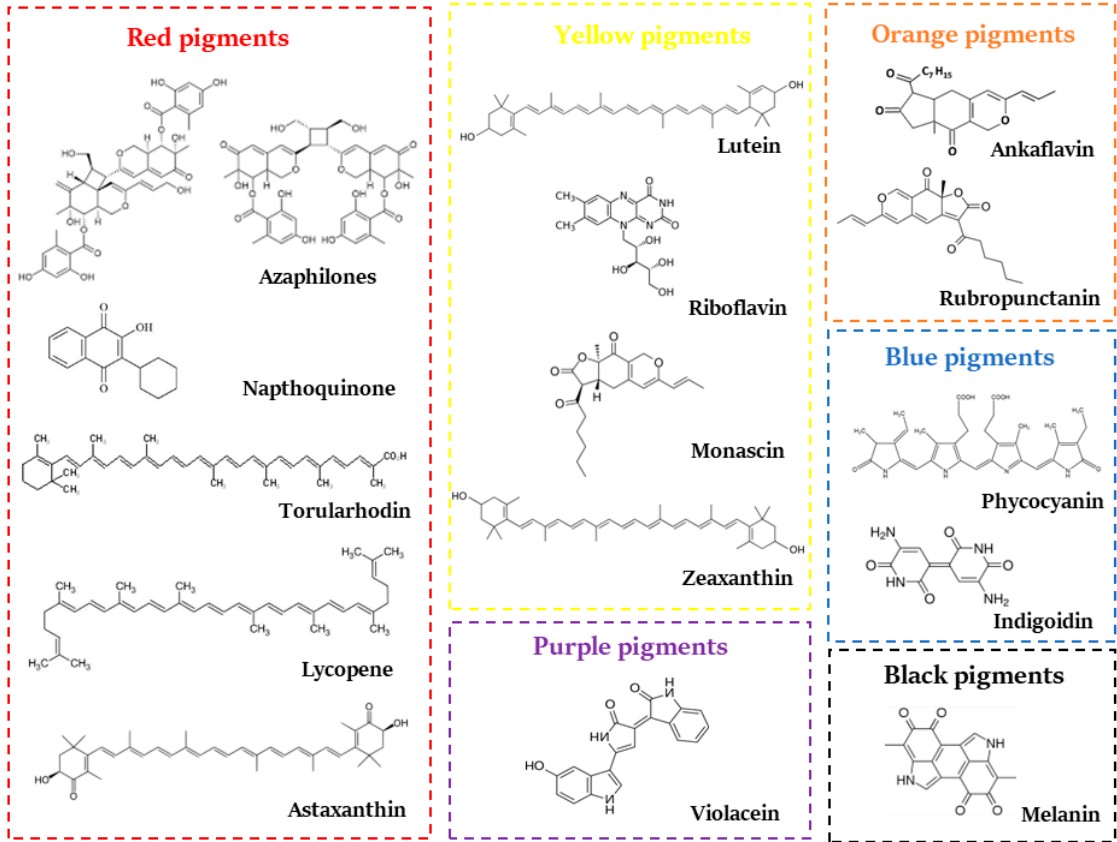

**Figure 1.** Category and structure of main representatives of microbial pigments.

## 3. Melanin Production via Fermentation

The commercialization of the melanin depends on the source that it is obtained while process viability in terms of cost and sustainability requires niche markets. Melanin that occurs in animals and plants is usually related to other compounds that render the whole downstream process complicated, due to the recalcitrance and complex structure of the extracted polymer. Synthetic melanin production is carried out via the oxidation of phenolic or indolic substrates using specific enzymes or chemicals. Although this process results in a high degree of melanin purity, it is cost intensive. Alternative melanin production via fermentation could offer process viability, especially when renewable waste streams are valorized. Fermentation media rich in yeast, i.e., 37 g/L, apart from being cost intensive, could complicate the downstream of melanin since some of the compounds contained in yeast extract can react with melanin precursors. Consequently, scientific research should focus on the formulation of fermentation media with readily assimilable nitrogen sources in the form of free amino nitrogen and simple carbon source preferentially derived from renewable waste streams [30]. When fermentation media are supplemented with melanin precursor molecules of different origin, specific melanin pigments can be produced due to the selective action of tyrosinases [30].

The most common microbial species that produce melanin are *Streptomyces*, *Pseudomonas*, *Rhizobium*, *Bacillus*, *Brevundimonas*, *Trichoderma*, *Shewanella*, *Aspergillus*, *Aeromonas* [31] and *Cryptococcus* [32]. Black fungi of the genera *Exophiala, Cladophialophora* and *Pseu-*

*dallescheria* have also been reported as potential melanin producers. They are found in highly polluted environments and have the ability to bind and degrade volatile aromatic hydrocarbons, including toluene, ethylbenzene and styrene. Some of these species can grow with these compounds as their sole source of carbon [33]. Table 2 presents the fermentation output of microorganisms when various waste and by-product streams have been employed for melanin production. Some commercial-based fermentation substrates as well as genetically modified strains have also been included.

**Table 2.** Microbial melanin production using various renewable resources based on literature-cited publications.

| Feedstock | Microbial Strain | Fermentation Time (Days) | Melanin (g/L) | Melanin Type | References |
|---|---|---|---|---|---|
| Renewable waste and byproduct streams | | | | | |
| Rice husk, rice flour and banana | *Streptomyces gresiorubens* DKR4 | | | | [34] |
| Carrot peel extract | *Aureobasidium pullulans* | 15 | 4.4 | | [35] |
| Wheat bran extract | *Auricularia auricula* | 5 | 0.519 | | [36] |
| Fruit waste extract | *Bacillus safensis* | | | | [37] |
| Marine residues | *Streptomyces roseochromogenes* ATCC 13400 | | 39.4 | | [38] |
| Pomegranate pulp | *Aspergillus carbonarius* M333 | 15 | | | [39] |
| Vegetable cabbage waste | *Pseudomonas* sp. | 48–72 h | | | [12] |
| Commercial-based fermentation media | | | | | |
| Dextrose, yeast extract and peptone | *Aspergillus fumigatus* AFGRD105 | | 0.007 | | [40] |
| Glucose, yeast extract and peptone | *Amorphotheca resinae* KUC3009 | 14 | 4.5 | eumelanin | [41] |
| Voguels salt water | *Hortaea werneckii* | 7 | 0.938 | | [42] |
| Amylodextrine, yeast extract and NaCl | *Streptomyces kathirae* | | 13.7 | | |
| Glucose, yeast extract and peptone | *Armillaria cepistipes* (Empa strain 655) | 161 | 27.98 | eumelanin | [5] |
| Genetically modified strains | | | | | |
| | *Bacillus thuringiensis* 4D11 | | | | [43] |
| | *S. kathirae* SC-1 | | 28.8 | | [44] |
| Caffeic acid | *E. coli* BL21(DE3) | 12 h | 0.17 | eumelanin | [45] |
| Glucose | *E. coli* | 120 h | 3.2 | eumelanin | [46] |
| Glycerol | *E. coli* W3110 | | 1.2 | catechol-melanin | [47] |

Scientists have developed several strategies to enhance the fermentation efficiency of melanin production. As it is depicted in Table 2, critical factors that have already been investigated to maximize concentration, yield, and productivity include temperature, pH, dissolved oxygen levels, inducers/precursors, and carbon to nitrogen ratios. It should be highlighted that the supplementation of the fermentation media with L-tyrosine favors melanin-polymer yields that resemble eumelanin properties. Melanin is produced as a secondary metabolite, often occurring in the stationary phase of fungal growth due to nutrient deficiencies of the medium.

### 3.1. Melanin Production Using Commercial-Based Fermentation Media

The Box–Behnken design in response surface methodology (RSM) has been reported as an important tool to optimize the process condition for melanin production. Media optimization led to a significant increase in melanin concentration (6.6 mg/L), which was stable after 5 days of fermentation when *A. fumigatus* AFGRD105 was used. In the same study, several commercial carbon sources (including dextrose, galactose, saccharose, mannitol, and sorbitol) as well as different concentrations of yeast extract and peptone (2.5–20 g/L) were evaluated. Dextrose was the best carbon source for both biomass and

melanin formation followed by galactose, saccharose, sorbitol and mannitol. Increasing concentrations of yeast extract as well as peptone up to 20 g/L led to increased melanin and biomass production [40]. In the study of Oh et al. (2020) [41], the ability of 102 different fungal strains to produce extracellular melanin was investigated. *Amorphotheca resinae* KUC3009 was chosen as the most efficient candidate. The fermentation media contained glucose as the carbon source as well as yeast extract and peptone as the nitrogen source. Melanin was mainly produced during the autolysis phase, reaching a concentration of 4.5 g/L after 14 days of fermentation. Based on structural analysis, it was demonstrated that the polymer was very similar to eumelanin, making it a good candidate for potential industrial-scale production. Various strains of *Hortaea werneckii* were screened in Voguels nutrient media, prepared with seawater to match their natural habitat. The strain that appeared to form the highest amount of melanin within 7 days of fermentation was *H. werneckii* AS1. The experiments were performed at a pH of 5–5.5, 30 °C, under both static and agitated cultures. The Plackett–Burman design increased melanin output by 1.23 fold. The variables of calcium chloride (1.125 g/L), trace element (0.25 mL/L), and culture volume (225 mL/500 mL) were optimized, leading to the maximum melanin yield of 0.938 g/L [42]. *Armillaria cepistipes* has been reported to produce enhanced amounts of eumelanin-type polymer. More specifically, a melanin concentration of 27.98 g/L was achieved within a 161-day fermentation period when a commercial-based nutrient medium consisting of glucose (1%) and peptone (1%)/yeast extract (0.1%) was used [5]. Marine debris (*Posidonia oceanica egagropili*) was investigated as the nutrient source for *Streptomyces roseochromogenes* ATCC 13400 to produce melanin. The formulation of the fermentation media involved various amounts of the marine residue combined with glucose, malt extract, and yeast extract. When 2.5 g/L of residue was applied to the nutrient medium, melanin was 7.4-fold higher compared to the control fermentation, with the highest concentration reaching up to 3.94 g/L [38]. In another study, *Streptomyces kathirae* (designated SC-1) was proven the most efficient melanin producer out of forty-five bacterial strains isolated from soil. Batch fermentation was optimized via a response surface method. More specifically, under the optimal culture conditions of 3.3 g/L amylodextrine, 37 g/L yeast extract, 5 g/L NaCl, 0.1 g/L $CaCl_2$, 54.4 μM $CuSO_4$ and a pH of 6, melanin reached the very high concentration of 13.7 g/L [48].

### 3.2. Low-Cost and Renewable Agri-Food Waste and Byproduct Streams

Inexpensive and abundant feedstock derived from effluents of the agro-industrial and food sector, from vegetables, fruits, cereals, etc., could lead to economically viable bioprocesses simultaneously deviating from the elevated environmental footprint. In the European Union (EU) alone, approximately 88 million tons of food are wasted at an annual basis, during all stages of production and consumption chain. This amount corresponds to ~173 kg of food waste per EU inhabitant/year [49] while it is projected to increase drastically by 2050 due to global population growth. Food waste is rich in valuable compounds such as carbohydrates, proteins, and lipids while their content depends on the origin or the production stage of food waste. So far, the most common method of disposal of food waste is landfilling. The global agri-food sector is estimated to generate approximately 5 billion tons of waste (in the form of peels, skins, leaves, seeds, pulps, or food products that do not meet specific quality criteria) [50] per year while the corresponding amounts that are generated within the EU have been reported equal to 76.5–102 million tons/year according to Eurostat [51,52]. Their high damaging effect on the environment is dictated by the fact that they are responsible for the annual emission of 3.3 billion tons of $CO_2$ [53]. Agricultural-derived waste is characterized by high chemical and biological oxygen demand [54], while their disposal has become a major challenge for the food industries due to the high costs that are required for their managment [55]. This source of waste contains phenolic and other compounds with a potential toxic effect that can lead to environmental degradation when their management is not properly addressed [22]. To date, agri-food waste and by-product streams have been used to produce biopigments, biopolymers, organic acid, biofuels,

animal feed and fertilizers. The availability of modern technologies has introduced new concepts, leading to the efficient utilization of agri-food by-products for the development of value-added products [56]. The thought behind the utilization of these by-products is to reuse them using sustainable technologies. Their value can be enhanced by turning them into value-added market chains that will boost local and global economies [57]. For instance, the extraction and/or production of intermediates that can be further integrated into the sectors of cosmetics, nutraceuticals, functional food, and bioenergy have been extensively reviewed [50].

### 3.3. Repurposing Agri-Food Waste via Melanin Production

There are very limited scientific studies that have so far investigated the production of microbial melanin employing renewable resources and bioprocessing. This is an ongoing and emerging strategy for sustainable and effective production of this fascinating and multifunctional biopolymer. Microbial melanin normally accumulates in the cell wall and/or it is excreted extracellularly when submerged fermentation strategies with certain C/N ratios are applied [21]. RSM was used to optimize melanogenesis in *Aureobasidium pullulans* NBRC 100716 when it was cultivated in carrot peel extract. Under optimum conditions—a pH of 5.9, 29.9 °C, 120 rpm, and 15 days of fermentation—the predicted value for total melanin concertation was determined equal to 4.4 g/L while intracellular and extracellular melanin production were respectively 2.4 g/L and 1.9 g/L. The aforementioned data of the experimental model were experimentally verified with very similar values [35]. Another cost-effective feedstock that was evaluated for melanin synthesis was wheat bran extract when the fungal strain *Auricularia auricula* was cultivated under shake flask fermentation. It was demonstrated that L-tyrosine and CuSO4 amounts significantly affected the tyrosinase activity and thus melanin production. The Box–Behnken design showed that an optimal melanin concentration (519.5 mg/L) could be achieved using 26.8% wheat bran extract, 1.6 g/L L-tyrosine and 0.1 g/L $CuSO_4$ after 5 days of fermentation. This study indicated that the valorization of wheat-bran-based fermentation media could lead to the development of an efficient strategy to prepare value-added melanin with potential to be industrially produced [36]. Apart from submerged fermentations, SSF using renewable solid matrices, i.e., apple, pomegranate, black carrot, and red beet pulps, has been evaluated for melanin production using the fungus *Aspergillus carbonarius* M333. SSF was investigated under different pH values (4.5–8.5), fermentation times (3–15 days), and initial particle sizes of waste pulp. The highest melanin production was obtained on pomegranate pulp at a pH of 6.5, with a particle size lower than 1.4 mm, during 15-day fermentation. The melanin was yellowish brown while SEM images showed that melanin was formed on the conidia of *A. carbonarius* [39]. SSF with *Streptomyces gresiorubens* DKR4 using residues such as rice husk, rice flour and varibanana peels has also been reported as an effective approach for melanin biosynthesis when optimum conditions—a pH of 7.5, a temperature of 40 °C and 50% initial moisture content—were applied [34]. The bacterial strain *Bacillus safensis* was studied for melanin production in fruit waste extracts. Optimum conditions—a pH of 6.84 and a temperature of 30.7 °C—resulted in the production of 96 g/L melanin. The produced melanin was found to have high photoprotective ability as well as radical and metal-chelating activity. All these properties make *B. safensis* a candidate for industrial applications [37].

### 3.4. Genetic Tools for Microbial Melanin Production

Genetic engineering methods target the overproduction of specific biometabolites by microorganisms or they can create novel recombinant strains. A first approach is the generation of melanin producers via the expression of specific genes that encode enzymes of tyrosinases family. In this technique, the tyrosinase gene is put under the control of an inducer in a replicative plasmid vector. The main disadvantages of this approach are the requirements for antibiotics (to prevent the growth of plasmid-less cells) and chemical inducers that increase the cost of melanin production and complicates subsequent

downstream processes [30]. It has been demonstrated that melanin synthesis by *Bacillus thuringiensis* is driven by the action of tyrosinase, which shows a catalytic action towards L-tyrosine to melanin, following the L-DOPA metabolic pathway. The authors cloned a tyrosinase-encoding gene (*mel*) from *B. thuringiensis* 4D11 employing a PCR method. *mel* was successfully expressed in recombinant *Escherichia coli*, leading to stable melanogenesis while melanin enhanced UV resistance of the latter [43]. Another study performed the cloning of the *melC* gene and its promoter P$_{skme}$l from the genomic DNA of *S. kathirae*. The protein sequence of tyrosinase showed 84% similarity with tyrosinase from *S. galbus*. The *melC* was introduced into *S. kathirae*, with the recombinant strain reaching the maximum melanin concentration of 28.8 g/L (2.1-fold higher than the wild strain) [44]. In the study of Ahn et al. (2019) [45], a novel microbial eumelanin-polymer (up to 0.17 g/L within 12 h) was produced using caffeic acid as a substrate via the whole-cell biotransformation of *E. coli* BL21(DE3) expressing feruloyl-CoA synthetase and enoyl-CoA hydratase/aldolase enzymes. The enzymatic bioconversion of caffeic acid resulted in protocatechualdehyde [45].

Another genetic engineering technique for enhanced melanin production is the random mutagenesis, which is rather restricted to natural melanogenic microorganisms [30]. *Pseudomonas putida* F6-TR and F6-HDO mutants were constructed using transposon (Tn5) mutagenesis to overproduce melanin under SSF. The mutant F6-TR showed 3.7-fold higher tyrosinase activity compared to the native strain when ferulic acid was involved. Mutant F6-HDO produced a red melanin pigment, which was 6-fold higher compared to the wild type while the biomass production was lower. Both mutants presented a higher survival rate compared to the wild strain when they were exposed to UV light (254 nm). Melanin produced by the F6-HDO mutant offered additional increased resistance towards H$_2$O$_2$ [58].

Metabolic engineering strategies, increasing the precursor supply, have also been used for maximum melanogenesis when simple sugar monomers are employed as the carbon source instead of L-tyrosine. In the study of Chávez-Béjar et al. (2013) [46], the Mut*melA* gene from *Rhizobium etli* was expressed in *E. coli* to encode an improved mutant tyrosinase. The engineered strain was able to produce up to 3.2 g/L of eumelanin-like pigment after 120 h of fermentation, utilizing glucose as the carbon source. A metabolically engineered *E. coli* strain was used to produce 1.2 g/L of catechol-melanin, fermenting 40 g/L of glycerol under batch bioreactor cultures. The gene Mut*melA* was integrated into the chromosome of *E. coli* W3110 trpD9923 that was modified via expression of genes encoding an enzymatic complex from *Pseudomonas aeruginosa* PAO1 [47].

## 4. Melanin Polymers

Among bio-based pigments, melanins show great scientific interest. Melanins are characterized as nitrogenous polymeric compounds [59], with the indole ring as the monomer. The name melanin has a rich history, as it comes from the ancient Greek word "melanos" that means "dark" in ancient Greek; its purpose was to describe the very dark pigment of the membranes of the eyes. Then the word "melanin" appeared, which was first used by Berzelius in 1840 to describe the black pigmentation in animals. Since then, the word "melanin" has been widely used to describe black or dark brown organic pigments [4]. Melanins are a mixture of macromolecules, exhibiting an irregular structure [21]. Melanins are produced by many types of organisms, from bacteria to mammals. In mammals, melanin is found in the skin, hair, eyes [2], *subsantia nigra* and the *locus coerruleus* of the brain [4], while it is found in the cell wall or as an extracellular polymer in cells. Melanin deposited in cell walls interacts with the chitin structure. A disruption in chitin metabolism leads to a release of melanin from the cell wall [32]. Melanin also occurs in the strains of many microbes, such as bacteria, fungi, and helminths [60].

Melanization occurs in several species of organisms when they are exposed to extreme environmental conditions as a survival strategy [2]. The evolution of species has led to their ability to regulate melanin in tissues, thus regulating color [61]. The different sources

of melanin production lead to heterogeneity in composition, color and function rendering challenging identification and categorization attempts [2,32].

### 4.1. Properties and Classification of Melanin

Melanin is insoluble in most solvents, and it shows high resistance to chemical degradation. It should be highlighted that no enzymes that could degrade the melanin molecule have been reported and thus it shows great stability [4]. Melanin is a heterogeneous polymer of high molecular weight (318.3 g/mol) [4] and it is formed by the oxidation of phenolic or indolic compounds and the subsequent polymerization of the resulting intermediates [2]. Despite the diverse origin of melanin, the basic physicochemical characteristics are left unaffected. Another special feature of this fascinating molecule is that it has the ability to be decolorized after contact with oxidizing agents such as $KMnO_4$, $NaOCl$ and $H_2O_2$ [4].

Melanin possesses remarkable and widely known properties—photoprotection, heavy metal blocking and free radical quenching. In addition, it enhances cell strength against heat and cold stress. The chemical characteristics, structure and extract multifunction of this complex molecule need to be specified in order to serve with high efficiency in value-added end uses [32]. The classification of melanin types is based on differences in its chemical composition. Eumelanin and allomelanin are characterized by a brown-black color, unlike pheomelanin, which is red or yellow. Pheomelanin appears to be located near the nucleus of the pigmented molecule, while eumelanin surrounds its surface [6]. In addition to the aforementioned categories, melanin can also be identified as neuromelanin, catechol melanin, DHN-melanin and pyomelanin [4]. Pyomelanin is found in the *coreruleus locus* of the brain and in catecholaminergic neurons [62]. Some researchers have classified pyomelanin as a subclass of allomelanin, whereas others recognize it as a distinguishable type [8].

### 4.2. Structural Diversity of Melanin

The chemical structure of melanin and its diversity have been a critical issue of investigation for many years [32]. Several methods have been used to identify and determine the structure of melanin (Figure 2), including thermogravimetric analysis (TGA), Raman stereoscopy, scanning electron microscopy (SEM), X-ray scattering, nuclear magnetic resonance (NMR), and Fourier-transform infrared spectroscopy (FT-IR) [62,63]. The main structure of melanin consists of 5,6-dihydroxyindole (DHI)) subunits attached at positions 2, 3, 4 and 7. Although different sources or extraction/purification methods of melanin show varying IR spectra, there are several characteristic bands, which can be trailed to identify the major functional groups, which are typical for melanin [61]. Indicatively, the FTIR spectra of melanin presents absorbance bands within 3600–3000 $cm^{-1}$ (attributed to stretching vibrations of -OH and -NH found in indolic and pyrrolic systems), 1650–1600 $cm^{-1}$ (attributed to aromatic C=C and C=O stretches of carboxylic function) and 1500–1400 $cm^{-1}$ [62].

The amount and distribution of melanin in the cell walls of microorganism show great variations. Electron microscopy studies have shown that melanin layers appear as granules. These layers present a pore size within 1–4 nm in diameter [60]. The pore size may increase either with increasing melanin amount or with increasing cell age.

Natural eumelanin pigment has several structural differences compared to synthetic eumelanin. The common feature of both eumelanins is the indole molecule, which forms the aromatic core of the molecule. However after investigation of double cross-polarization and proton-assisted insensitive nuclear cross-polarization (PAIN-CP) NMR, four magnetically different indole pairs of 13 C-15 N were found [8,60,64]. It has been reported that synthetic melanin has a high content of alkylated pyrrole and phenol derivatives and a lower content of indole, benzene and pyridine when compared to melanin derived from three different *Drosophila melanogaster* strains [62,63]. Melanin presents a high resistance to thermal degradation while decomposition temperatures vary (500–1000 °C) due to different

origins of melanin, i.e., synthetic, bacterial derived, or fungal derived. Synthetic melanins tend to show higher thermal resistance compared to their natural counterparts [62].

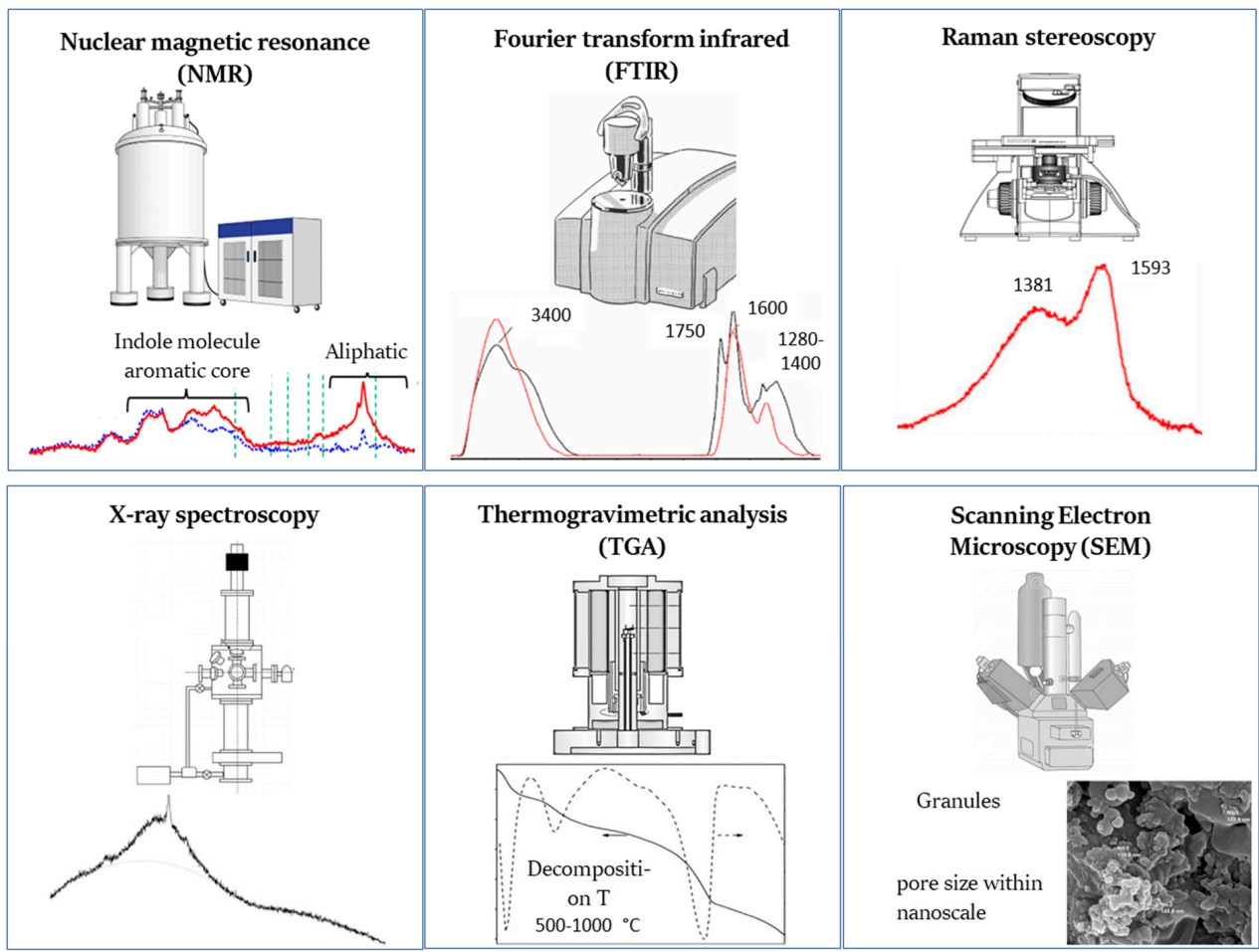

**Figure 2.** Most applied methods for melanin characterization including nuclear magnetic resonance (NMR), Fourier-transform infrared (FTIR), Raman stereoscopy [9], X-ray spectroscopy [65], thermogravimetric analysis (TGA) [62], and scanning electron microscopy (SEM) [42].

### 4.3. Biosynthesis of Melanin from Microbes

Melanogenesis is a process that primarily involves the enzymes of tyrosinase or laccase to oxidize polyphenolic compounds (i.e., L-tyrosine, L-DOPA, and catechols) that are available as substrates. Tyrosinase and/or laccase have been reported as melanin inducers and thus they have been related to the melanin biosynthesis process in several bacterial strains. In fungi, tyrosinase, laccase and polyketide synthase normally oxidize aromatic compounds related to specific melanin biosynthesis routes [31]. The tyrosinases are monooxygenases with a dinuclear copper catalytic center and they catalyze monophenols and catechols to generate ortho-quinone derivatives. The lacasses, which are also involved in melanin synthesis, have been found in bacteria, fungi, and plants and they are also copper-dependent oxidoreductases [30].

Microbial melanin is biosynthesized either through the metabolic pathway of DOPA (tyrosine as the precursor) or the DHN pathway (malonyl coenzyme A as the precursor) (Figure 3) [2]. The main enzymes that are involved in these metabolic pathways include polyphenol oxidases, or tyrosinases, laccases and catechol oxidases with copper ions playing a crucial role in their activity. More specifically, copper ions contribute to the coordination of molecular oxygen and the orientation of the reducing substrate to achieve catalysis [32]. Hydroxylated aromatic compounds that are accumulated due to enzymatic

imbalances during the catabolic process [8], lead to the production of alternative melanin types [2].

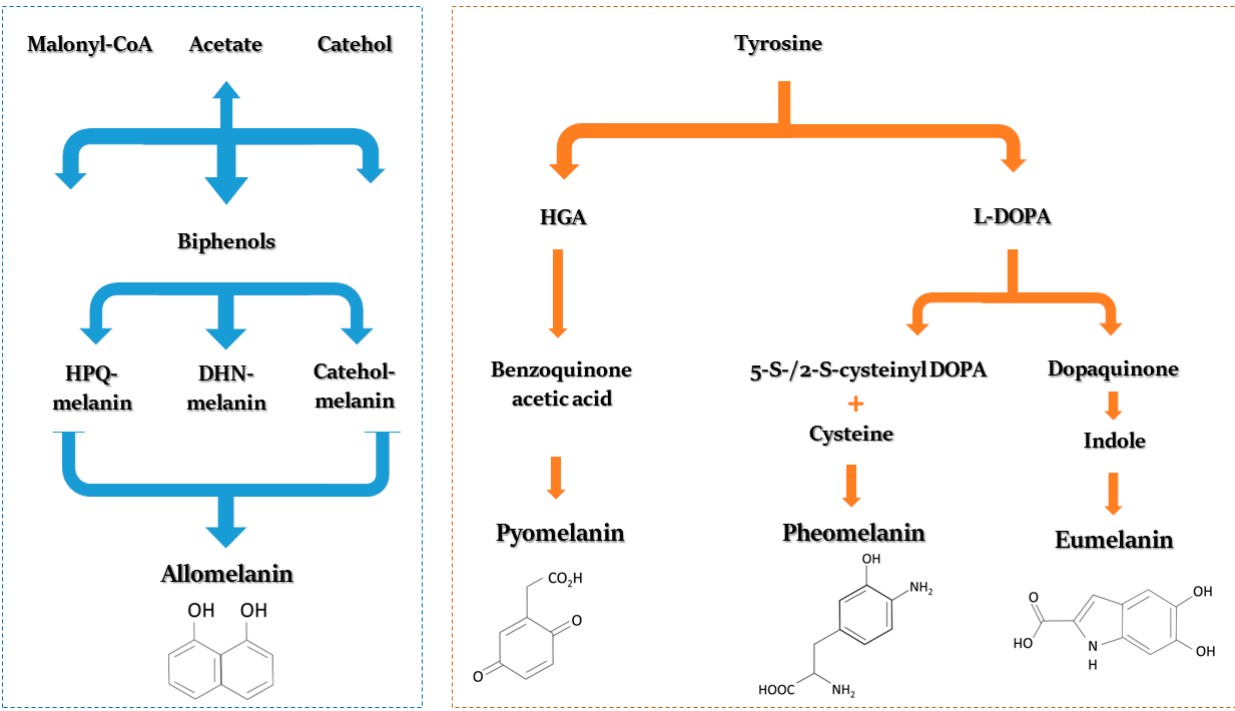

**Figure 3.** Simplified metabolic pathways for microbial melanin biosynthesis in bacteria and fungi [8,31].

Eumelanin and pheomelanin are generated after the oxidative polymerization of tyrosine to produce L-3,4-dihydroxyphenylalanine (L-DOPA), which then oxidizes spontaneously or facilitated by catalysts to DOPAquinone (precursor compound of melanin) [6]. The latter is either cyclized to DOPAchrome (idole) to further yield eumelanin via polymerization reactions or it is cyrsteinylated to cysteinyl-DOPA(2-S/5-S) to further produce pheomelainn. More specifically, cysteinyl-DOPA reacts with cysteine to form the polymer of pheomelanin [7]. The L-DOPA pathway has been described in yeast strains i.e., *Cryptococcus neoformans*, while it resembles melanogenesis in certain bacterial strains and animals [31].

Allomelanin is the less studied and the most heterogeneous class of melanin polymer. The class of allomelanin includes 1,8-dihydroxynaphthalene (DHN-melanin), 3,4-dihydroxyphenylacetate (HPQ), and catechol-melanin. Allomelanins are derived from the oxidation of N-free precursors, that is manolyl-CoA, catechol, and acetate, to respectively produce HPQ-melanin, catechol-melanin, and DHN-melanin via oxidative polymerization of precursors molecules (γ-glutaminyl-3,4-dihydroxybenzene, 1,3,6,8-tetrahydroxynaphthalene (THN) and catechol). The most common type of allomelanin is 1,8-DHN-melanin. The catalyzation occurs from a polyketide synthase that eventually makes di-DHN, 1,8-DHN. [4,8,9,31,62]. In cases where precursors containing amino groups in their structure are involved, L-DOPA is produced. The DHN-melanin metabolic pathway typically occurs in bacteria, thus it has also been identified in *Ascomycetes* such as the fungi *Sporothrix schenckii*, *Magnaporthe grisea* and *Neurospora crassa* [31].

Pyomelanin is an end product derived from the tyrosine catabolism. Homogentisic acid (HGA) is produced after the catalysis of 4-hydroxyphenylpyruvate employing the enzyme 4-hydroxyphenylpyruvic acid dioxygenase. The subsequent autoxidation of HGA leads to benzoquinone acetic acid formation and finally results in pyomelanin [8].

## 5. Property-Dependent Applications of Melanin

Melanin constitutes a complex polymer that can be exploited in multiple biotechnological, medicinal, and environmental sectors due to its unique properties of radiopro-

tection, antioxidant activity, chelation of heavy metals and absorption of organic compounds [30,63,66]. Figure 4 depicts sources of melanin and value-added properties and applications.

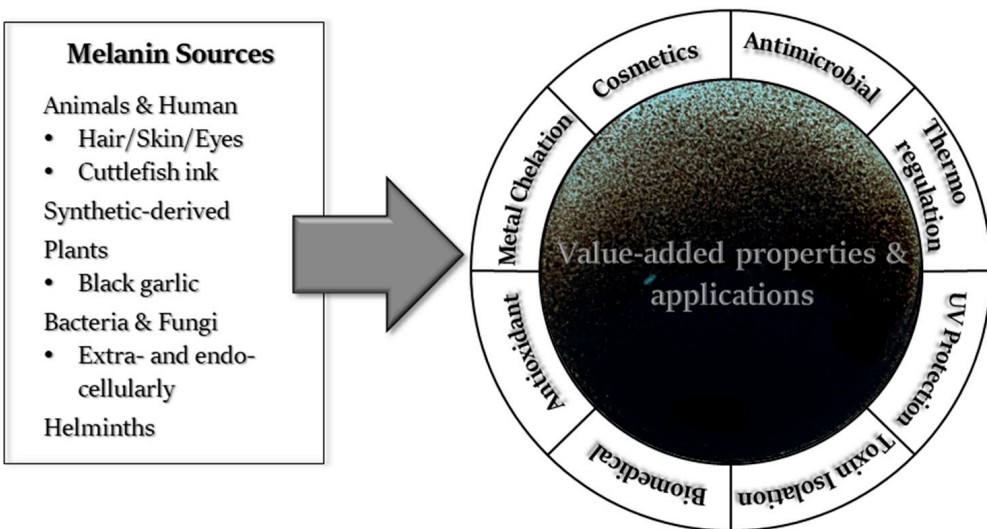

**Figure 4.** Sources of melanin and value-added properties and applications.

*5.1. Absorption and Binding Capacities of Melanin and Related Properties*

The radiation absorption capacity of melanin at a wide range of wavelengths within the UV and visible electromagnetic spectrum can be extremely useful in environmental applications, such as hybrid photovoltaics and photodetectors, promoting green energy [61]. In fact, it is proven that melanin has the capacity to diffuse almost 99% of the UV radiation that it absorbs through non-radiative media and also eliminates the reactive oxygen species (ROS) [31,61]. The high UV absorption potential of melanin derives from the absorption and scattering of light by molecules that are coupled in the melanin polymer [62]. The presence of melanin in the black yeast *E. lecanii-corni* significantly increased its resistance to elevated doses of UV-C and γ-radiation. The potential of melanin to absorb UV radiation is well demonstrated while the protective action of melanin against ionizing radiation is not well studied and understood [26]. The fungus *Aureobasidium pullulans* is well known for its capability to resist against gamma and ionizing forms of radiation. This makes it a 'sunscreen' of nature, which can provide multivarious applications in the cosmetic industry, including sun protection [2]. Additionally, *Pseudomonas* sp. SSA has been used for cosmetic purposes to its ability to produce extracellular melanin with enhanced UV protection properties.

Melanin can be efficiently used as a biosorbent for removal of heavy metals from aquatic territories. The integration of melanin with polymers such as polycaprolactone or polyurethane can result in the removal of up to 94% Pb in aqueous systems [2]. Liquid effluents that possess a heavy load of metal ions have been conventionally remediated via chemical precipitation, reverse osmosis, ion-exchange, and filtration techniques, thus these methods are efficient only in metal concentrations higher than 100 ppm. However, lower concentrations than that are also toxic and dangerous for human life. To add, it has been reported that treatment with $Ca(OH)_2$ and flocculants of acidic waster containing high levels of uranium in the Osamu Utsumi mine area are extremely cost intensive (from USD 200,000 to USD 250,000 at a monthly basis) [67,68]. Melanin produced via green technology, was efficiently used for the biosorption of uranium from aqueous solution. Melanin demonstrated good uptake over a wide pH range while uranium was fastly removed after 2 h of contact [68].

Melanogenic fungi and bacteria can be used to isolate toxins from contaminated environments. Melanin contributes to their survival in adverse environments while their ability to catabolize organic volatile compounds is enhanced. Thus, this makes them ideal

microorganisms for the development of combined strategies that are based in bioprocessing and biocatalysis [69]. The aforementioned render melanin capable to successfully find application in bioremediation of contaminated soils and as biocatalysts in air biofilters [33].

### 5.2. Antimicrobial Activity

The increasing and extensive use of commercial antimicrobial agents in several applications have resulted in the widespread antimicrobial resistance of pathogens and thus scientific research seeks for novel antimicrobial agents that can deal with this alarming threat. Human infections from pathogenic bacterial and fungal strains are a problem that affects the food industry, and medical equipment since ineffective prevention of infections can lead to serious health consequences. Synthetic antimicrobial agents are characterized by high costs that burden their high-scale production while the vast majority are not environmentally friendly. Melanin has been proposed as an effective antimicrobial agent. Melanin produced from the mushroom *Schizophyllum commune* has been shown to have a significant contribution to reducing infection by drug-resistant pathogenic bacteria and antifungal activity against fungi of the genus *Trichophyton* [31]. Further studies have demonstrated that melanin isolated from *Lachnum* YM30 [70] and the saprophytic fungus *Exidia nigricans* displays enhanced antibacterial activity against pathogens such as *Listeria monocytogenes, Bacillus mecillus, Staphylococcus aureus, Salmonella typhi, Vibrio parahaemolyticus* and *Escherichia coli*. Recently, many other studies have demonstrated the antibacterial activity of microbial melanin against various pathogens, and thus it can be useful in various applications such as in biomedicine and pathology fields to prevent infections [31].

### 5.3. Antioxidant Activity

The molecule of melanin is still being investigated for its suitability to be used in food packaging formulations. However, studies have shown that melanin could be used to enrich fatty products such as pork lard, to further delay or prevent lipid oxidation. This effect is likely to be due to the antioxidant activity of melanin and its ability to neutralize free radicals [66]. The demand for natural antioxidants is constantly increasing especially after numerous scientific indications that synthetic compounds, i.e., butylated hydroxytoluene or butylated hydroxyanisole, could possess toxicity [71]. Melanin has attracted scientific interest due to its remarkable antioxidant activity, which is attributed to the chemical arrangement of the melanin molecule. Melanin contains both reducing and oxidizing moieties, through which it can bind free oxygen through electron exchange. The most suitable structures of melanin that can offer enhanced antioxidant capacity are eumelanins and pheomelanins [4]. Melanin reacts more similarly to a one-dimensional semiconductor in which free radicals are trapped by its protons [61]. In fact, it has been shown that microbial melanin can neutralize up to 80.9% of free radicals, which makes it much more effective than the action of synthetic melanin [31]. The oxidation state of melanin and other melanin-like pigments resembles the oxidation state of basic and characteristic natural antioxidants during their biosynthesis, i.e., the polyphenols and quinones [72]. Eumelanin produced from a recombinant *E. coli* strain was applied to produce hydrogels for soft contact lens dyeing presenting exceptional dyeing capacity simultaneously providing antibacterial and antioxidant activity, as well as a higher water-content rate compared to synthetic melanin-based contact lens [45].

### 5.4. The Case of Food Packaging Formulation

The production of biobased films for food packaging applications could effectively address several end-of-life management issues that are related to the conventional (non-degradable and fossil-based) plastics. In recent years, there has been a great interest in the production of biofilms from biopolymers. Approximately 400 million tons of plastics is produced annually, while almost 40% is used for food packaging. The preference for fossil-based packaging material arises due to their low cost compared to their biobased counterparts. However, the footprint they leave on the environment has led to a search for

greener solutions [73–75]. Biofilms present enhanced physical and functional properties as a result of the enclosed bioactive compounds [76]. The antioxidant activity of melanin may find appealing applications in the field of health. It has been shown that the melanin pigment can counteract the pathogenic effect of hydrazane in the liver. Hydrazane is carcinogenic, so preventing its action is associated with better liver health [4].

### 5.4.1. Biopolymers for Food Packaging Materials

Normally, biopolymers present high gas barrier capacities, are recyclable, biodegradable, biocompatible and non-toxic, and thus they have been reported as ideal materials for food packaging applications. Cellulose is the most abundant natural polymer, with a wide range of applications due to its biocompatibility and its great film-forming capacity. Cellulose is also a great support carrier for antioxidant and antimicrobial compounds [77]. Chitosan is the second most plentiful polymer in nature, and it derives from crustacean chitin [78]. Chitosan has been reported to be suitable for edible film preparation with antibacterial activity mainly against Gram-negative bacteria. Neat chitosan-based films present a low mechanical response with poor barrier potential. The involvement of plasticizing agents and/or nanoparticles that act as crosslinking agents, i.e., ZnO and AgNPs, can improve mechanical profile of final formulations [79]. Sodium alginate is a polysaccharide that can be used in food packaging due to its colloidal properties, low cost, easy handling and functionalization, and biodegradability. However, it presents a poor physicochemical profile that makes it difficult to be involved in effective food packaging formulation. Supplementation of alginate films with polymers and nano-fillers can produce packaging with enhanced mechanical and barrier properties [76]. The study of Motelica et al. [80] showed that ZnO–essential oil–alginate films could extend the shelf life of cheese, offering antimicrobial properties and protection from UV light. The same scientific group demonstrated that Ag(nanoparticles)–essential oil–alginate films could offer satisfying antimicrobial protection, color, surface texture, and softness of cheese for up 14 days [81]. Poly(lactic acid) (PLA) is a bioplastic polyester that derives from the lactide ring polymerization. PLA presents low price and high bioavailability. It is characterized as a thermoplastic material with high rigidity and clarity that could be compared to polystyrene or poly(ethylene terephthalate) (PET). PLA can increase the tensile strength potential of biofilms simultaneously offering high composability [75,82].

### 5.4.2. Melanin Enhanced Biofilms

Melanin is a nanostructured polymer that could effectively be used in food packaging formulations. Melanin exhibit high stability to various food components such as glucose, sucrose, potassium ascorbate and ascorbic acid [4]. The antioxidant effect of melanin in active packaging films has been evaluated, while results showed the capacity of melanin to prevent food oxidation [76]. Additionally, pork lard was packaged using melanin-enriched membranes, resulting in the prevention of rotting due to fat rancidity. After enriching the packaging with melanin, free radicals were neutralized and the degree of oxidative rancidity of the product was reduced [66]. Studies have shown that increasing the amount of melanin nanoparticles in food packaging membranes is positively and linearly related to their antioxidant activity [74]. The UV-protective property of melanin is also a feature that makes it ideal for use in food packaging. In the study conducted by Bang et al. [83], it was demonstrated that the production of melanin-based composite films, applied in potatoes, gave the product a high degree of protection against UV radiation, especially UV-B radiation. Melanin also improved the mechanical properties of the packaging as well as the vapor and oxygen permeability and its color. Thus, to avoid spoilage of potatoes, which is a problem during storage and distribution, melanin-enriched films are a very good eco-alternative [83].

Roy et al. [74] studied carrageenan-based films, and they demonstrated that the incorporation of melanin into the films led to effective UV blocking and increased their water vapor permeability (WVP). This phenomenon was attributed to the formation of a discontin-

uous phase between the matrix of the carrageenan molecule and the melanin nanoparticles. Decreased values of WVP have been reported for films derived from blends of melanin and gelatin while increasing melanin concentration into the films led to decreasing WVP [84]. The effect of melanin on film vapor permeability values was extensively studied by Shankar et al. [84]. Specifically, they enriched gelatin with melanin nanoparticles (~100 nm) at concentrations within 0–1.0%. The resulting films were uniform, due to compatibility between the melanin–gelatin complex. This research showed that the melanin-enriched films had significantly improved vapor permeability and mechanical properties as well as thermal stability. The results varied depending on the melanin concentration, but their properties were better compared to control samples (without melanin) [84]. Carrageenan-based films fortified with melanin nanoparticles (40–160 nm) presented improved thermostability and UV absorption. The mechanical strength of the membranes increased significantly until the addition of 0.1% melanin. Finally, it was shown that the addition of melanin developed hydrophobicity and high water barrier ability of the film [74].

The performance of CMC membranes supplemented with melanin from the mushroom *Agaricus bisporus* and carvacrol, was evaluated. The developed biofilms exhibited resistance to bacterial strains including *Candida albicans, Escherichia coli* and *Staphylococcus aureus*. Also, melanin seemed to improve the antioxidant capacity, WVP, mechanical properties of the biofilms as well as their ability to protect food products against radiation. In this case, melanin did not affect the transparency of the films, even though it affected their color [73].

Various concentrations of fungal melanin (0.025%, 0.05% and 0.2%) were added to PLA membranes and mechanical, antioxidant, antimicrobial and UV vapor barrier properties were determined. The lowest concentration of melanin improved the barrier and mechanical properties of the films. The presence of melanin appeared to enhance the antimicrobial activity of films against *Pseudomonas putida*, *Enterococcus faecalis* and *Pseudomonas aeruginosa* as well as their antioxidant activity, while the UV protective properties were slightly increased. The opacity of the films was decreased with increasing melanin concentrations [75].

Melanin nanoparticles extracted from *Pseudomonas* sp. was incorporated in polyhydroxybutyric acid-based films. The produced films were homogeneous, flexible and smooth while they presented good antimicrobial activity against gram positive and negative bacteria as well as high thermostability at temperatures up to 282 °C. The addition of *Pseudomonas* sp. melanin to food packaging will protect products from bacterial contamination. Finally, the absorption of free radicals increased with increasing melanin concentration from 20 and 120 mg/L [85].

## 6. Conclusions and Future Perspectives

The production of melanin via fermentation is strain dependent, while the components of the medium as well as temperature, pH values and the supplementation with precursors should be thoroughly investigated for maximum melanin production. Microbial melanin polymers can be applied in the multivarious sectors of food, pharmaceuticals, textiles, cosmetics, electronics and environment science-related applications. Genetic and metabolic engineering strategies could effectively modify the expression of native genes in natural melanin producers to achieve product overexpression as well as generate innovative melanogenic strains with enhanced melanogenesis capacity. Certain aspects of biochemistry and genetics related to melanin metabolic pathways from bacteria and fungi have facilitated direct manipulation, especially when simple carbon sources are employed. The development of fermentation processes based on agricultural and agro-based industrial wastes could facilitate the transition towards a low carbo-economy simultaneously eliminating disposal issues, elevated production cost of melanin and fulfilling sustainability, renewability and biocompatibility requirements of environmental circularity. Agri-food residues could be an excellent feedstock to produce bio-based and bioactive compounds, especially when bioprocessing is employed. The implementation of innovative and green

strategies to revalorize waste enables their re-introduction into value-added supply chains and thus transition from linear processes to circular bioeconomy concepts.

The market demand for melanin is exponentially increasing due to the unique properties of this multidimensional molecule. Even though melanin presents enhanced potential as a functional molecule, commercialization is still limited. This is mainly attributed to its high biodiversity that is related to difficulties in obtaining tailor-made properties and a final product of standard quality and performance. The search for non-pathogenic melanogenic microorganisms is a great challenge that must be addressed, as far as final applications demanding safe building blocks. Techno-economic evaluation of the melanin production process should be taken into consideration, given that the product concentration is normally within the mg/L scale, especially when wild-type strains are used, even when they are cultivated in synthetic media, without any inhibition from inhibitory compounds. Future investigation should indispensably include proper and detailed mapping of melanin properties to target and sustain emerging sectors—from photoprotection (dermato-cosmetics) to radioprotection, bioremediation, food preservation, and biomedicine. The future of melanin-based products lies in the ability to increase melanin production yield at a low cost while simultaneously employing green processing.

**Author Contributions:** Conceptualization, E.T. (Erminta Tsouko) and D.S.; software, E.T. (Erminta Tsoukoand) and E.T. (Eirini Tolia); investigation, E.T. (Erminta Tsouko) and E.T. (Eirini Tolia); writing—original draft preparation, E.T. (Erminta Tsouko) and E.T. (Eirini Tolia); writing—review and editing, E.T. (Erminta Tsouko) and D.S.; project administration, D.S.; All authors have read and agreed to the published version of the manuscript.

**Funding:** This research received no external funding.

**Institutional Review Board Statement:** Not applicable.

**Informed Consent Statement:** Not applicable.

**Data Availability Statement:** Not applicable.

**Acknowledgments:** This research was funded by the project "Infrastructure of Microbiome Applications in Food Systems-FOODBIOMES" (MIS 5047291), which is implemented under the Action "Regional Excellence in R&D Infrastructures", funded by the Operational Programme "Competitiveness, Entrepreneurship and Innovation" (NSRF 2014-2020) and co-financed by Greece and the European Union (European Regional Development Fund).

**Conflicts of Interest:** The authors declare no conflict of interest.

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
