# Peer review of "Microbial Melanin: Renewable Feedstock and Emerging Applications in Food-Related Systems"

_sustainability, doi:10.3390/su15097516_

Round 1

Reviewer 1 Report

This article is about the most important natural pigments - melanin, produced by various organisms from microbes-to-plants-and-mammals. Melanins have been found to possess great properties such as radioprotective and antioxidant activity, heavy metal chelation and absorption of organic com-pounds.

The article has been well developed and presents issues related to the formation and production of melanins by microorganisms, taking into account genetic aspects. The material contained in the article refers to current research trends. The article systematizes knowledge in this field, it is helpful in understanding the mechanisms of formation at the cell level. The only caveat is that the authors should look at the cited literature, specifically the list of literature, in order to develop it in accordance with the requirements of the journal, e.g. item 11, 12, 14, 18, 25, 27, 50, 88.

Author Response

Comments are supplied in a pdf file

Reviewer 2 Report

In this work a review is made where the recent advances achieved in the production of melanin are presented, emphasizing the valorization of renewable resources, the applications of food packaging formulations and the biosynthetic pathways for the production of melanin from fungal and bacterial strains.

The abstract and literature is appropriate.

In the introduction is duplicated, delete (44 a 58).

Authors need to be more clear about the benefits of their work.

The authors have done a serious job of summarizing and presenting important points in this topic. It is desirable to indicate more clearly the further prospects for the development of microbial melanin and its derivatives.

Please improve the quality of the Figure 1.

In Chapter 4.2. some structural methods are mentioned, here their understanding would be simpler if figures of the materials mentioned in the chapter were placed.

It seems the paper was thoroughly revised and looks refined; however, there are still some typographical errors in the manuscript.

The paper is in good shape and can be considered after author revised these minor changes.

Author Response

Comments are supplied in a pdf file

Reviewer 3 Report

Comments 

The manuscript entitledMicrobial melanin: renewable feedstock and emerging applications in food-related systems” have focused on natural pigment melanin produced by various microorganisms, plants and mammals. Numerous forms of melanin can be synthesized through various pathways including DOPA-metabolic pathway and/or DHN-pathway, melanin also have the high bioactive potential. The formation of biobased films for food packaging applications reinforced with melanin nanoparticles, constitutes a market segment of high interest due to environmental and societal concerns on end-of-life management of conventional plastics, gradual decrease in fossil resources, sustainability issues and higher performance. The manuscript is written well, however, the following comments need to be addressed: 

Line 38-39: Rewrite sentence 

Line 59: Add reference (citation) column in the table  

Line 187-190: Rewrite sentence 

Line 196-200: Avoid writing direct abstract, add their findings in your review 

Line 267-273: Avoid writing long paragraphs without citation: add citation. 

Line 387-389: Rewrite sentences depicting phenomena of melanin clearly 

Line 552-556: Modify language. 

Line 603-608: Rewrite sentence 

General comments: Modify language of the manuscript 

Author Response

Comments are supplied in a pdf file

Reviewer 4 Report

The article "Microbial melanin: renewable feedstock and emerging applications in food-related systems" describes the state of the art in the melanin obtaining / applications domains. It is a valuable study that can be published after authors address the following problems:

I suggest adding neuromelanin as keyword too.

Introduction: “There are four main categories of melanin” and after that authors name four of them on row 38 and a fifth on row 40. Please see doi: 10.1021/jacs.0c12322 “Generally, melanin is classified into five types” and correct the description.

Then again at rows 380-381 one can read “Depending on its chemical composition, melanin is classified into three basic types: eumelanin, pheomelanin and allomelanin” !!! What are your thoughts about this inconsistency?

As a suggestion, if some melanin types are considered more important or are more used by the organisms, authors can simply state: “from the five melanin types, the most encounter in the nature are ....” or “the most important from applications point of view are the following types....”

Rows 32-44 are repeating at rows 44-56.

Some statements are inexact while authors do cite the right source: e.g. rows 68-70 – please see again reference 5 (may I ask how much melanin has white hair? There is a reason why the reference article mentions black hair. In addition, there are many more melanin sources listed there beside those 3 mentioned in the present review). A review must cover the subject in an extensive manner.

Again statements like “both plant- and synthetic-based ones are produced/extracted with low efficiency” from rows 97-98 are inexact as long the subject can be any pigment. While authors are trying to argument that microbial pigments can be produced cheaper and all year around, while plants only grow seasonally and synthetic pigments require expensive initial setup, the delivered statement from rows 97-98 is simply not true.

While the objective of this review is clear, authors should indicate also the review methodology (keywords used, databases consulted, years interval considered, other criteria). For review criteria please see PRISMA for example.

The English language needs some polishing for style and typos (e.g. row 28 extra dot; row 73 “,.”; row 124 “ex{Citation}treme”; row 639 “Pseudomonas” should be italicized; ref 88 all caps)

In section 5.4.1. for chitosan and alginate based packaging few more examples are welcome, like doi: 10.3390/foods9121801, doi: 10.3390/pharmaceutics13071020 or doi: 10.3390/nano11092377.

A section “Trends and perspectives” before conclusions would increase the value of the article.

Author Response

Comments are supplied in a pdf file

Round 2

Reviewer 4 Report

The authors have responded to my comments and have addressed all my concerns, substantially improving the manuscript, therefore, I suggest publishing the paper in the current form.